# Single-Cell RNA Sequencing of a Postmenopausal Normal Breast Tissue Identifies Multiple Cell Types That Contribute to Breast Cancer

**DOI:** 10.3390/cancers12123639

**Published:** 2020-12-04

**Authors:** Sen Peng, Lora L. Hebert, Jennifer M. Eschbacher, Suwon Kim

**Affiliations:** 1Cancer and Cell Biology Division, Translational Genomics Research Institute, Phoenix, AZ 85004, USA; speng@tgen.org; 2Department of Surgery, St. Joseph’s Hospital, Dignity Health, Phoenix, AZ 85013, USA; lora.hebert@dignityhealth.org (L.L.H.); jennifer.eschbacher@dignityhealth.org (J.M.E.); 3Surgical Breast Oncology Division, University of Arizona Cancer Center-Phoenix, Phoenix, AZ 85004, USA; 4Department of Neuropathology, Barrow Neurological Institute, Dignity Health, Phoenix, AZ 85013, USA; 5Department of Basic Medical Sciences, University of Arizona College of Medicine-Phoenix, Phoenix, AZ 85004, USA

**Keywords:** single-cell RNA sequencing, normal breast, cluster analysis, GSVA, mammary epithelial cells, cytokeratin expression, mammary fibroblasts, TCGA breast cancer dataset: breast cancer, triple-negative breast cancer

## Abstract

**Simple Summary:**

The human body is composed of multiple cell types that form structures and carry out the functions of specific tissues. The human breast is mainly known for the milk ducts organized by epithelial cells, but also contains many other cell types of little-known identity. In this study, we employed the single-cell sequencing technology to ascertain the various cell types present in the normal breast. The results showed 10 distinct cell types that included three epithelial and other novel cell types. The gene signatures of five cell types (three epithelial, one fibroblast subset, and immune cells) matched to the gene expression profiles of >85% breast tumors cataloged in The Cancer Gene Atlas dataset, suggesting their significant contribution to breast cancer. These findings provide a framework for the better mapping of the cellular composition in the breast and its relationship to breast disease.

**Abstract:**

The human breast is composed of diverse cell types. Studies have delineated mammary epithelial cells, but the other cell types in the breast have scarcely been characterized. In order to gain insight into the cellular composition of the tissue, we performed droplet-mediated RNA sequencing of 3193 single cells isolated from a postmenopausal breast tissue without enriching for epithelial cells. Unbiased clustering analysis identified 10 distinct cell clusters, seven of which were nonepithelial devoid of cytokeratin expression. The remaining three cell clusters expressed cytokeratins (CKs), representing breast epithelial cells; Cluster 2 and Cluster 7 cells expressed luminal and basal CKs, respectively, whereas Cluster 9 cells expressed both luminal and basal CKs, as well as other CKs of unknown specificity. To assess which cell type(s) potentially contributes to breast cancer, we used the differential gene expression signature of each cell cluster to derive gene set variation analysis (GSVA) scores and classified breast tumors in The Cancer Gene Atlas (TGGA) dataset (*n* = 1100) by assigning the highest GSVA scoring cell cluster number for each tumor. The results showed that five clusters (Clusters 2, 3, 7, 8, and 9) could categorize >85% of breast tumors collectively. Notably, Cluster 2 (luminal epithelial) and Cluster 3 (fibroblast) tumors were equally prevalent in the luminal breast cancer subtypes, whereas Cluster 7 (basal epithelial) and Cluster 9 (other epithelial) tumors were present primarily in the triple-negative breast cancer (TNBC) subtype. Cluster 8 (immune) tumors were present in all subtypes, indicating that immune cells may contribute to breast cancer regardless of the subtypes. Cluster 9 tumors were significantly associated with poor patient survival in TNBC, suggesting that this epithelial cell type may give rise to an aggressive TNBC subset.

## 1. Introduction

Human tissues contain morphologically distinguishable structures composed of multiple cell types. Histological sections of breast tissues typically show the mammary ducts embedded in the collagenous tissue matrix that houses fibroblasts, adipocytes, blood vessels, immune cells, and other stroma components. Many studies have focused on characterizing the mammary ducts, not only because they deliver the tissue function of producing milk after childbirth, but also because the majority of breast tumors originate from epithelial cells that constitute the mammary ducts [1,2]. Two histologically distinct epithelial cell types constitute the mammary ducts: luminal cells that form tightly sealed cellular lining of the lumen and basal cells that form a contractile cell layer that interphases the luminal cell layer and the basement membrane attached to the tissue matrix [3]. Basal cells are also referred as myoepithelial cells for their characteristic expression of both epithelial and muscle-related markers [4]. Luminal and basal epithelial cell types are differentiated by their unique sets of cytokeratin (CK) expression such as luminal CK8/18 vs. basal CK5/14/17 [3,4], and histological staining for these CKs has been used to detect disseminated tumor cells in the lymph nodes, blood, and distant metastatic sites [5,6].

In recent years, single-cell RNA sequencing (scRNAseq) has emerged as a powerful tool to delineate the composition of different cell types and/or “cell states” in a given tissue using differentially expressed gene sets as a parameter [7]. scRNAseq has been applied to evaluate tumor heterogeneity, tumor-associated immune cells, and therapy-induced tumor evolution in breast cancer [8,9,10,11]. These studies have reported that adjacent normal tissues were analyzed in parallel but used mainly to subtract out normal components from tumors such that information on the various cell types that are present in the “normal” breast tissue has been difficult to obtain.

Mammary epithelial cell types have been better characterized compared to the others in the normal breast. Studies have utilized cell markers including EpCAM (epithelial cell adhesion molecule) and Cd49f (α6 integrin) to classify epithelial cells isolated mainly from reduction mammoplasty samples from premenopausal women [12,13]. The consensus built around mammary epithelial cell types using the two cell surface markers, EpCAM and Cd49f, is that three major cell types are present in adult breast tissues, referred as mature luminal (EpCAM^+^CD49f^−^), luminal progenitor (EpCAM^+^CD49f^+^), and basal (EpCAM^−^CD49f^hi^) cells [12,13,14,15,16]. Using differentiation markers, transcription factors, and/or gene expression sets, studies have further subclassified the three epithelial cell types to 7–12 cell types or “cell states,” demonstrating the plasticity of breast epithelial cells likely depending on multiple factors such as genetics, age, the hormonal status, and the individual’s health and environment [14,15,17]. Consistent with this idea, Lim et al. reported that individuals harboring a *BRCA1* germline mutation had the luminal progenitor (EpCAM^+^CD49f^+^) population aberrantly expanded in pre-disease breast tissues, suggesting that luminal progenitor cells may give rise to tumors in *BRCA1*-mutated hereditary breast cancer [12]. More recently, Nguyen et al. characterized normal breast epithelial cells at single-cell resolution using scRNAseq [13]. Similar to Lim et al., the study used antibody-mediated sorting to isolate EpCAM^+^ epithelial cells followed by CD49f^lo^ vs. CD49^hi^ separation before proceeding to scRNAseq [13]. Nguyen et al. also reported three major epithelial cell types referred to as mature luminal (L2), luminal progenitor (L1), and basal (B) cells [13]. Although these two studies were in agreement regarding the three major breast epithelial cell types, some ambiguity exists as to the discerning features of each epithelial subtype. For example, Lim et al. classified basal cells as EpCAM-negative, whereas basal cells in Nguyen et al. were among EpCAM^+^ presorted cells [12,13]. This could be in part due to variable EpCAM immunoreactivity, as well as varied single-cell isolation protocols.

In order to better define epithelial and other cell types present in the breast, we employed droplet-based scRNAseq (10× Genomics Chromium, Pleasanton, CA, USA) of cells isolated from a normal breast tissue without presorting for EpCAM or CD49f surface markers, thus presenting an inclusive landscape of the diverse cell types that constitute the breast. Additionally, the breast tissue used in this study was a single postmenopausal mastectomy sample, different from the pooled premenopausal reduction mammoplasty samples used in the previous studies. Accordingly, our study results offer a simpler landscape of the breast without the influence of hormones or individual variabilities. The results showed the presence of three epithelial cell types evidenced by differential CK and *EPCAM* expression. In addition, we identified seven distinct nonepithelial cell types that included smooth muscle, endothelial, fibroblast, and immune cells. We compared the gene signature of each cell type to the gene expression profiles of breast tumors in The Cancer Genome Atlas (TCGA) dataset and found that five cell types significantly contributed to breast cancer.

## 2. Results

### 2.1. Single-Cell Isolation and Single-Cell RNA Sequencing (scRNAseq) of Normal Breast Cells

The workflow of scRNAseq is summarized in Figure 1A. The normal breast tissue obtained from a mastectomy patient sample was apportioned into several sections, one of which was formalin-fixed and paraffin embedded for histological examination. Hematoxylin and eosin staining visualized various cellular structures including the mammary ducts surrounded by stroma and prominent adipocytes (Figure 1B). Another ~0.3 g portion of tissue was used to isolate viable single cells. In brief, we modified a mouse mammary organoid preparation protocol and used mechanical dissociation followed by collagenase IV digestion as an initial step (mammary organoid pictures in Figure 1C) [18]. It was notable that fat, nerve bundles, and blood cells were visibly excluded during differential centrifugation washing and cell strainer steps. Organoid-containing tissue fractions were then sequentially treated with dispase II and trypsin to yield a single-cell suspension (Figure 1D). Single-cell sizes ranged approximately from 0.5 to 15 μm in diameter (Figure 1D).

From the single-cell suspension, 3352 cells were captured by nanodrop oil droplets in the 10X Genomics kit, amounting to ~750,000 barcode indexing. We sequenced 170 million reads in total, accounting for an average ~50,000 reads per cell. The sequence data of the single-cell library showed that 20,345 genes were expressed collectively, while each cell cluster was classified by differential expression of an average 1290 genes according to the Cell Ranger analysis pipeline. These results indicated that breast tissue expressed more than 20,000 genes collectively, indicating that the tissue as a whole expressed the majority of the estimated ~25,000 total number of genes in the human genome [19]. Comparatively, each cell type isolated from the breast tissue was defined by a relatively small ~6% fraction of the total number of genes expressed in the tissue (1290 out of 20,345 genes).

We also carried out exome sequencing of the bulk tissue as part of other study [20], which identified no obvious gene mutations associated with hereditary cancers. RNA sequencing of the bulk tissue showed the gene expression profile that closely matched to a commercially available universal breast tissue control [20], which supported the designation of our tissue sample as a “normal breast”.

### 2.2. Single-Cell Transcriptomic Landscape of the Breast Shows 10 Distinct Cell Clusters

We used the Seurat toolkit that separated breast cells into 10 clusters based on differentially expressed genes (DEGs) [21]. The data were visualized using a t-distributed stochastic neighbor embedding (t-SNE) plot in two-dimensional projection space. Seurat function “FindAllMarkers” was used to identify a set of signature marker genes for each cluster compared to all other cells [21].

The cluster numbers were assigned from the largest cell number (Cluster 0, *n* = 769) to the smallest (Cluster 9, *n* = 31) (Figure 2A). The number of cells and characteristic genes in each cluster are summarized in Table 1, and the 100 DEGs in each cluster are listed in Appendix A. In brief, the largest single-cell constituent Cluster 0 cells were smooth muscle cells with muscle-specific gene expression including *MYL9* (myosin light chain 9) and *MYH11* (myosin heavy chain 11) [22,23,24]. Cluster 1 cells were endothelial cells expressing *EMCN* (endomucin), *ENG* (endoglin), and *SELE* (selectin E), all of which have shown endothelial-specific expression [25]. Cluster 2 represented luminal epithelial cells with high expression of luminal cytokeratin (CK) genes, *KRT7/8/18/19*, whereas Cluster 7 represented basal epithelial cells expressing basal CK genes, *KRT5/14/17* [3,4]. Cluster 7 cells also expressed *ACTA2* (smooth muscle actin 2) and *TAGLN* (transgelin), consistent with the characteristics of basal/myoepithelial cells [26,27]. Cluster 3 represented fibroblasts that expressed extracellular matrix genes including several collagen genes such as *COL1A*, *COL3A*, and *COL6A* [28], likely responsible for laying down the collagenous tissue matrix (Figure 2B and Table 1; see Section 3).

Additionally, we identified several potentially novel cell types. Cluster 4 cells appeared to be an endothelial cell subset expressing endothelial-specific genes such as *EDN1* (endothelin 1) and *ESAM* (endothelial specific antigen marker) [25], but evidently different from Cluster 1 endothelial cells (Table 1 and Figure 2A,B; see Section 3). Cluster 6 cells appeared to be related to Cluster 3 fibroblasts, considering the t-SNE similarity relationship (Figure 2A), as well as the heatmap showing overlapping gene expression (Figure 2B). Both Cluster 3 and 6 cells expressed *ALDH1A1*, a fibroblast marker [12]. Other genes that were common between Clusters 3 and 6 included *COL1A1/A2* and *COL6A2*, suggesting that both cell types are likely to contribute to collagen deposition of the breast matrix. How they are different from each other is presently unclear (see Section 3). Differentiation trajectories generated using Monocle placed Cluster 1 preceding Cluster 4 in the contiguous pseudo-timeline, as well as Cluster 6 preceding Cluster 3, suggesting different cell types than “cell states” (Appendix A). However, these require functional verifications.

Intriguingly, Cluster 5 cells expressed genes supposedly expressed in several cell types of different lineage such as adipocytes, macrophages, smooth muscle, and fibroblasts. Cluster 5 cells also expressed epithelial–mesenchymal transition (EMT) markers (Table 1), suggesting an undetermined cell lineage. Expression of multiple lineage genes has been described for bone marrow-derived mesenchymal stem cells [29]. Thus, we propose that Cluster 5 cells represent a mesenchymal progenitor population with potential to differentiate into multiple cell types. Supportive of this idea, the Monocle pseudotemporal analysis showed that Cluster 5 cells mapped at the “root” in a contiguous differentiation trajectory to Cluster 0 (smooth muscle) in one branch, to Cluster 3 and 6 (fibroblast subsets) in another branch, and to Cluster 1 and 4 (endothelial subsets) in the third branch (Appendix A). However, functional assays would be needed to demonstrate if Cluster 5 cells have mesenchymal progenitor properties.

Cluster 8 cells were immune cells, specifically expressing cytotoxic T-cell markers such as *CD3D*, *CD8A*, and *GZMK* (granzyme K) (Table 1) [30]. Although underrepresentation of the immune cell types could be explained by the loss of the cell population during our single-cell isolation protocol as others have also reported [31], it was unclear as to why CD8 T cells preferentially survived the preparation steps among other immune cells such as macrophages, for example.

Lastly, Cluster 9 cells were the third epithelial cell type expressing both luminal and basal CKs, as well as other CKs of unknown cell type specificity, including *KRT15/16/23/81* (Table 1, see below and Section 3).

### 2.3. EpCAM and ITGA6 Expression in Breast Single-Cell Clusters

As several studies have previously characterized breast epithelial cells using the two cell surface markers, EpCAM and CD49f [17], we evaluated expression levels of the two genes, *EPCAM* and *ITGA6* (encoding CD49f/alpha 6 integrin), in our scRNAseq data. The results showed that *EPCAM* was expressed only in the epithelial cell types, Cluster 2 (luminal), Cluster 7 (basal), and Cluster 9 (other) (Figure 3A), supportive of the rationale using EpCAM as an epithelial cell marker. Cluster 9 cells expressed the highest amount of *EPCAM* with a relative median value that was 2.7-fold and 9.6-fold higher compared to Cluster 2 and Cluster 7 cells, respectively (Table 2). Cluster 7 basal cells expressed the lowest amounts of *EPCAM* compared to Cluster 2 and 9 epithelial cells, consistent with other studies. However, considering the large standard deviations within each cell cluster (Figure 3A and Table 2), caution may be warranted when classifying epithelial cells according to the amounts of EpCAM alone.

In contrast to *EPCAM*, whose expression was restricted to epithelial cells, *ITGA6* was expressed in endothelial cells, as well as in epithelial cells. In fact, Cluster 1 and 4 endothelial lineage cells expressed higher levels of *ITGA6*, compared to the others (Figure 3B, Table 2). Among epithelial cells, Cluster 2 luminal cells did not express *ITGA6* (Figure 3B, Table 2), whereas Cluster 7 (basal) and Cluster 9 (other) cells expressed comparable amounts of *ITGA6* (*p* > 0.1, Figure 2, Table 2). This integrin expression supported the identity of Cluster 7 cells as basal, which enables attachment to basement membrane [32]. By the same token, Cluster 9 cells may be located in the basal layer of the duct as Cluster 7 cells.

These *EPCAM* and *ITGA6* expression levels in the three epithelial cell clusters closely resemble the previous studies in which breast epithelial cells were classified by the EpCAM and Cd49f cell surface expression: Cluster 2 correlated with EpCAM^hi^Cd49f^−^ “mature luminal”, Cluster 7 with EpCAM^−/low^Cd49f^hi^ “basal,” and Cluster 9 with EpCAM^+^Cd49f^+^ “luminal progenitors” [12]. The characteristics of Cluster 9 cells are further described (see below).

### 2.4. Cytokeratin Gene Expression in the Three Breast Epithelial Cell Types

To gain insight into the three distinct epithelial cell types, we compared the relative transcript levels of cytokeratin (CK) genes between Cluster 2, Cluster 7, and Cluster 9 cells. CKs form intermediate filaments that provide structural support for the epithelia, and unique sets of CK expression are related to specific cell types and lineage differentiation [33]. CK filaments are typically formed by heteromeric pairings of type I and type II CKs. For example, CK 8/18 and CK 7/19 pairs are called major primary and secondary simple keratins, respectively [33]. Cluster 2 cells expressed genes encoding such keratin pairs, *KRT7*, *KRT8*, *KRT18*, and *KRT19*, consistent with the designation “luminal” epithelial cells (Figure 4A). In comparison, Cluster 7 cells expressed *KRT5*, *KRT14*, and *KRT17*, instead of *KRT18* and *KRT19* (Figure 4B). The CK 5/14 pairs are found in the basal layers of the epithelia that are in contact with the basement membrane, supportive of the designation of Cluster 7 cells as “basal/myoepithelial” cells. CK 10 (encoded by *KRT10* expressed in both Cluster 2 and 7 cells) has been shown to be expressed in the intermediate cell layers between the basal and luminal layers in some epithelia, so-called “supra basal” or “uppermost” layers [33]. Thus, *KRT10* expression in both Cluster 2 and Cluster 7 suggest that both luminal and basal cell types could constitute intermediate layers in the mammary ducts (Figure 4A,B).

Cluster 9 cells expressed all CKs expressed in Cluster 2 and Cluster 7 cells, as well as additional CKs encoded by *KRT6B*, *KRT15*, *KRT16*, *KRT23*, and *KRT81* (Figure 4C,D). Comparison of the average individual *KRT* gene expression among Cluster 2, 7, or 9 cells showed significantly different expression levels of the *KRT* genes except for *KRT8* and *KRT10*, which were expressed in all three cell types (Figure 4E). Thus, according to *KRT* expressions, Cluster 9 cells did not appear to be committed to the luminal lineage per se. Supportive of this, the t-SNE similarity relationship analysis placed Cluster 9 at an equidistance from Cluster 2 and Cluster 7 (Figure 2A), and the heat map showed that Cluster 9 cells expressed the DEGs overlapping with both Cluster 2 and Cluster 7 gene sets (Figure 2B). Collectively, these results suggested that Cluster 9 cells were significantly different from Cluster 2 (luminal) and Cluster 7 (basal) cells, representing a third epithelial cell type in the breast. Moreover, Cluster 9 cells expressed other *KRT*s of unknown lineage specificity, which may suggest an undetermined fate. Additionally, Monocle-generated pseudotemporal differentiation ordering showed Cluster 9 at the “root” contiguous to Cluster 7 on one arm and to Cluster 2 to another arm in the trajectory (Figure 4F). Interestingly, Cluster 2 definitively branched into two separate clusters (Figure 4F), suggesting that Cluster 2 may in fact consist of three different cell types or “cell states”, including the luminal progenitor cell population as described in Nguyen et al. [13]. Taken together, we propose that Cluster 9 cells may represent “bipotent” epithelial cells, rather than a luminal progenitor cell type (see Section 3).

### 2.5. Five Cell Cluster Signatures Representing 86% of Breast Tumors in TCGA Dataset

It has been established that breast cancers mainly emerge from mammary epithelial cells [2], but the scope of other cell types contributing to breast cancer has been less characterized. We investigated this by comparing the single cell cluster DEG signatures to the gene expression profiles of 1100 breast tumors in The Cancer Genome Atlas (TCGA) breast cancer dataset. We performed gene set variation analysis (GSVA) [34], generating 10 GSVA scores per tumor. GSVA scores ranged from −1.0 to +1.0, representing low to high concordance related to the enrichment of the DEGs. Each tumor was then classified with the cell cluster number with the highest GSVA score, as it indicated that the tumor gene expression closely matched to the gene signature of the specific single cell cluster.

We counted the number of tumors assigned to each cell cluster number and found that the five cell types, Clusters 2, 3, 7, 8, and 9, could classify 946 out of 1100 tumors, collectively accounting for 86% of breast tumors in TCGA dataset (Figure 5A). The three epithelial cell type signatures classified 40.8% of tumors (Cluster 2, 22.6%; Cluster 7, 8.9%; Cluster 9, 9.3%; see Figure 5A and inset table), while the stroma and immune cell signatures collectively classified additional 45.2% of tumors (Cluster 3, 22.0%; Cluster 8, 23.2%; see Figure 5A and inset table). These results demonstrated that stroma and immune cells are among the major cell types that contribute significantly to breast cancer in addition to epithelial cells.

While epithelial cell gene signatures were expected to match the gene expression profiles of breast tumors, it was somewhat unexpected that the stroma or immune cell signature could classify just as many breast tumors as the epithelial cell gene signatures. We asked whether “stroma” or “immune” tumors were due to the prominent presence of stroma or immune “content” in epithelial tumors. In this case, the GSVA scores for Cluster 3 (stroma) or Cluster 8 (immune) may be similar in value to an epithelial cell cluster GSVA score. We evaluated whether the assigned cell cluster GSVA scores were significantly different from the other cell cluster GSVA scores, within each designated cluster tumor type. The box plot results showed that the average GSVA scores for the Cluster 2 signature was significantly higher than any other cluster GSVA scores in Cluster 2-designated tumors (Figure 5B, top left panel). Similarly, “Cluster 3 tumors” had significantly higher Cluster 3 GSVA scores, compared to any other cluster scores including Cluster 2 (luminal epithelial) or Cluster 6 (the second highest GSVA scores closer to Cluster 3 scores (* *p* < 0.0001, Figure 5B, top right panel,). These indicated that Cluster 3 tumors are not simply Cluster 2 epithelial tumors with high fibroblast content. Likewise, “immune” tumors had the Cluster 8 GSVA scores were significantly higher compared to other cluster scores (Figure 5B, bottom middle panel). These results indicated that the cluster designation according to the highest GSVA score per tumor was meaningful, representing the gene expression profile closely matching to the designated single-cell cluster.

### 2.6. Single-Cell Cluster Signatures Correlating with the Breast Cancer Subtypes

In order to examine whether any single-cell cluster signatures correlated with the molecular subtype of breast tumors, we first performed unsupervised clustering using the normalized gene expression of key subtype genes as described previously (Figure 6A, dendrogram) [35]. We then labeled each tumor according to the breast cancer subtype information extracted from the dataset (Figure 6A, top panel color map). The results showed that unsupervised clustering closely matched the subtype designation based on the status of the estrogen receptor (ER) and the HER2/neu receptor (HER2); ER^−^HER2^+^ subtype tumors mapped to the far left of the heat map (red), next to triple-negative tumors (green), followed by ER^+^HER2^+^ tumors (luminal B, dark blue), and ER^+^HER2^−^ tumors (luminal A, purple). To compare the receptor status with the receptor gene expression levels, we generated the heat maps for *ERBB2* (the gene encoding HER2), *ESR1* (the gene encoding ER), and *PGR* (the gene encoding progesterone receptor, PR) expression (Figure 6A, bottom three panels). Consistent with the subtype designation, elevated *ERBB2* expression was found in HER2^+^ tumors (Figure 6A, third panel, yellow). Concordantly, elevated *ESR1* expression was enriched in ER^+^ breast tumors (Figure 6A, fourth panel, yellow), but many expressed relatively low levels of *ESR1* (Figure 6A, fourth panel, blue vs. yellow). In particular, there were no discernable *ESR1* expression differences between ER^−^HER2^+^ and ER^+^HER2^+^ tumors (Figure 6A, top panel, red vs. dark blue), suggesting some ambiguity in the ER^+^ designation among HER2^+^ tumors, potentially accounting for differences between ER protein and RNA levels. *PGR* expression was mostly restricted to ER^+^ breast tumors (Figure 6A, last panel, yellow and blue).

Next, we generated a color map of TCGA breast tumors using the single-cell cluster signatures. The results showed that triple-negative breast cancer (TNBC) tumors (green; Figure 6A, top panel) were enriched for Clusters 7, 8, and 9 (green, blue, purple, Figure 6A second panel), whereas ER^+^ tumors matched primarily to Cluster 2 and 3 (orange, yellow orange; Figure 6A, second panel).

We then separated the TCGA tumors by the subtypes of ER^+^HER2^−^ (luminal A), ER^+^HER2^+^ (luminal B), HER2^+^ER^−^ (HER2), and triple-negative (TNBC) and counted tumors of designated cell cluster numbers according to the highest GSVA scores. The results showed that Cluster 2-, 3-, or 8-designated tumors were prevalent in ER^+^ tumors (Figure 6B, first two bars), whereas TNBC tumors consisted of Cluster 7, 8, or 9 designations (Figure 6B, last bar). The epithelial cell cluster signature separation between ER^+^ (Cluster 2) and TNBC (Cluster 7, 9) was striking, supportive of the paradigm that the two subtypes of breast cancers may have originated from different cell types: ER^+^ tumors arising from luminal cells (Cluster 2) and TNBC tumors arising from basal cells (Cluster 7). In addition to 21% of TNBC tumors classified as Cluster 7 basal epithelial tumors, 31% of TNBC tumors were closely matched to the Cluster 9 “other” epithelial cell signature (Figure 6B,C), suggesting that Cluster 9 cells contributed to a significant portion of TNBC (see Section 3).

In ER^+^ tumors, 32% of tumors matched to the Cluster 2 (luminal) signature and another 32% tumors matched to the Cluster 3 (fibroblast) signature (Figure 6B,C). Although our GSVA showed that Cluster 3 scores were significantly different from that of Cluster 2, it could not be discerned whether these were tumors with a high content of stroma/fibroblasts or that potentially originated from fibroblasts. Nonetheless, the exclusive presence of Cluster 3 fibroblast tumors in ER^+^ tumors departs from the current paradigm that cancer-associated fibroblasts (CAFs) contribute to hormone-negative breast cancer progression [36,37,38] (see Section 3).

In contrast to the Cluster 2 and 3 signatures prevalent specifically in ER^+^ tumors and Cluster 7 and 9 in TNBC tumors, both sets of cell cluster signatures (Clusters 2, 3, 7, and 9) were found in the ER-negative HER2^+^ subtype (Figure 6B,C), attesting to the heterogeneity of HER2^+^ tumors as reported previously [39].

The Cluster 8 (immune) signature constituted a significant percentage of tumors in all molecular subtypes (Figure 6B, blue), indicating that immune cells may contribute to breast cancer regardless of the subtype. TNBC tumors had the highest 43% matching to the Cluster 8 immune cell signature compared to 22–29% tumors in other subtype tumors (Figure 6B,C), supportive of the previous observations that TNBC tumors are associated with infiltrating immune cells [40,41].

Since our single-cell gene signatures were derived from a postmenopausal sample, we performed a separate analysis using the postmenopausal tumor sample data only (*n* = 564) and found the same distribution of Clusters 2, 3, 7, 8, and 9 as with the all-tumor sample data (Appendix A). These findings indicated that the gene signatures of the five cell types derived from a postmenopausal sample can represent the majority of breast tumors regardless of the patient menopausal status.

### 2.7. Poor Patient Survival in Cluster 9 TNBC Tumors

In order to evaluate whether any of the single-cell cluster signatures were associated with patient survival, we used Kaplan–Meier analysis using TCGA dataset. The results showed no significant differences in survival outcomes between patients with tumors assigned to Clusters 2, 3, 7, 8, or 9 (Figure 7A). We next examined whether any specific cell cluster types were associated with patient survival in the respective molecular subtypes. The comparison among Clusters 2, 3, and 8 prevalent in the ER^+^HER2^−^ or ER^+^HER2^+^ subtype did not show statistically significant differences in patient survival (Figure 7B,C), and neither did HER2^+^ tumors assigned to Cluster 2, 3, 7, 8, or 9 (Figure 7D). In contrast, TNBC tumors assigned to Cluster 9 (other epithelial cells) were significantly associated with poor prognosis (*p* = 0.043, log-rank test for trend, Figure 7E) compared to Cluster 7 (basal) or Cluster 8 (immune) tumors. Cluster 9 tumors were associated with 72% reduced overall survival in patients (hazard ratio, 0.2863; 95% confidence interval, 0.094 to 0.873), compared to Cluster 7 (basal) tumors. Taken together, these data suggested that an aggressive TNBC subset may originate from Cluster 9 “other” epithelial cells.

## 3. Discussions

### 3.1. Inclusive Single-Cell Landscape of the Normal Breast

Single-cell transcriptome analyses have enabled the assessment of cellular heterogeneity in normal and tumor tissues. While it is a powerful tool to obtain a snapshot of molecular details at single-cell resolution, the results have varied to some degree depending on how cells were prepared in each lab. In addition, the plasticity of mammary tissue during development, hormonal status, pregnancy, and lactation, as well as individual-to-individual variations, has added to the complexity of data interpretation [15,17,42,43].

Previous single-cell studies of normal mammary tissues mainly focused on epithelial cells by sorting for cell surface markers such as EpCAM prior to scRNAseq [13,44,45]. While the antibody-mediated sorting steps enable enriching for the target cell population, these could potentially introduce antibody-dependent bias toward selective cell types. Moreover, as it has become evident that mammary stroma in the tissue microenvironment plays a critical role in breast cancer progression and in therapy response [36,46], delineation of nonepithelial cell types in the breast bears significant importance. We presented here single-cell RNAseq data of a normal breast tissue without antibody-mediated presorting, thus providing a single-cell landscape of the breast inclusive of the diverse cell types. We acknowledge the limitation of our study that it is derived from a single tissue sample and that it requires validation. Nevertheless, we found that the results were instructive and may serve as a framework for the future detail mapping of the cell types in the breast.

### 3.2. Three Epithelial Cell Types (Clusterd 2, 7, and 9)

Our study of a single postmenopausal tissue sample offers a simple version of the breast epithelial landscape composed of three epithelial cell types. While the mature luminal (Cluster 2) and basal/myoepithelial (Cluster 7) cell type designations were definitive with their respective cell-type-specific cytokeratins and concordant gene expressions including *EPCAM* and *ITGA6*, the identity of the third epithelial (Cluster 9) cell type was somewhat ambiguous. We considered that Cluster 9 cells may represent the cell type referred to as “luminal progenitors” with EpCAM^+^Cd49f^+^ marker expression in the previous studies [12,13]. These cells were thought to be committed to luminal lineage because of the presence of luminal cytokeratin 8/18, but with diminished or no expression of basal cytokeratin 14 [12,13]. Cluster 9 cells, however, expressed *KRT14* along with other basal and luminal *KRT*s. Luminal progenitors were shown to express high levels of *MUC1* [47], but Cluster 9 cells did not express *MUC1*. The t-SNE plot showed that Cluster 9 cells separated from Cluster 2 and Cluster 7 cells, suggesting that Cluster 9 cells may not be committed to either lineage. Supportive of this idea, the pseudotemporal analysis placed Cluster 9 cells at the root between Cluster 7 on one arm and Cluster 2 on the other arm of the trajectory timeline (Figure 4F). Moreover, the Cluster 9 signature was found mainly in TNBC tumors, as was the Cluster 7 basal cell signature, potentially suggesting that Cluster 9 cells may be related to Cluster 7 basal cells rather than to Cluster 2 luminal cells, whose gene signature was mainly found in the luminal subtype of breast cancer. Another noticeable difference was that Cluster 9 cells represented the rarest epithelial cell type, only constituting 1% of total single cells, whereas Cluster 2 luminal and Cluster 7 basal cells were 14% and 5% of total single cells, respectively (Table 1). Compared to the scarcity of Cluster 9 cells we observed in our study, others have reported that luminal progenitors made up a large proportion of epithelial cells comparable to or larger than basal or mature luminal cell populations [12,13]. Our psudotemporal analysis showed that Cluster 2 luminal cells further branched to two separate trajectory lines (Figure 4F), resembling luminal progenitor cells and two luminal cell states described in Nguyen et al. [13]. Thus, Cluster 2 cells appeared to feature a large population of epithelial cells including luminal progenitors and other luminal cell states. Taken together, we advocate that Cluster 9 cells may represent bipotent progenitors rather than luminal progenitors, the least common epithelial cell type with a capacity to renew both the luminal and basal layers of the duct [14,47,48]. This may be consistent with the current understanding of bipotent mammary stem cells that exist and coordinate mammary ductal homeostasis, demonstrated by spectacular multispectral in situ lineage tracing, as well as by computational pseudotemporal lineage analyses [48,49].

Of note, our results showed that the epithelial cell type distribution ratio was 14:5:1 for luminal, basal, and other epithelial cell types. It is tempting to speculate that this ratio approximates the prevalence of the breast tumor subtypes: 70% luminal (Cluster 2) to 30% others (Cluster 7 and 9) [50]. It would be informative to determine if this ratio is reflective of a typical biological composition of postmenopausal breast tissues.

### 3.3. Nonepithelial Cells in the Breast That Do Not Contribute to Breast Cancer

The largest population of cells (Cluster 0) we isolated from the breast tissue sample were vascular smooth muscle cells expressing the specific class II myosin complex components, myosin heavy chain encoded by *MYH11* and light chain encoded by *MYL9,* as well as *MYLK*, myosin light chain kinase, whose major function is to provide vascular contractility [23,24]. *MYL9* has also been implicated in vascular remodeling during injury and aging [22]. Cluster 1 cells, the second largest cell population, were endothelial cells. Thus, Cluster 0 and Cluster 1 together constituted ~50% of our single-cell population isolated from the breast tissue, indicating the abundant presence of blood vessels in the normal breast. Whether these represent the cellular composition of a typical breast tissue or are due to our single-cell isolation method will need investigation. Cluster 4 shared some of the genes expressed in Cluster 1 endothelial cells, but also expressed specific endothelial markers such as endothelin and epherin B2, whose expression has been associated with arterial endothelial cell lineage [51,52]. The cellular location and identity of such an endothelial cell type could be obtained by tissue section staining.

Cluster 5 appeared to be closely related to Cluster 0 smooth muscle cells with the t-SNE plot showing a close proximity between the two cell types. However, Cluster 5 expressed genes specific to many different lineage cells including fibroblasts, adipocytes, and even macrophages. They also expressed EMT markers such as *SNAI2* and *TGFB1*, suggesting that these cells may be of an undetermined lineage and/or in a transitional state. The pseudotemporal differentiation trajectory ordering showed Cluster 5 cells at the “root” that branches to smooth muscle cells (Cluster 0), endothelial cells (Cluster 1 and 4), and fibroblasts (Cluster 3 and 6), suggesting that Cluster 5 may represent mesenchymal progenitors. However, the functional identification of Cluster 5 will require further investigation. The gene signatures of smooth muscle (Cluster 0), endothelial (Cluster 1 and 4), and unknown lineage (Cluster 5) cells did not match to TCGA breast tumors, suggesting that these cells may not directly contribute to breast cancer.

### 3.4. Fibroblasts (Cluster 3 and 6) and Cancer-Associated Fibroblasts (CAFs)

Stroma fibroblasts play a significant key role in maintaining homeostasis of the tissue and fulfill the function of laying down the extracellular matrix rich in collagen fibers and others [28]. Although fibroblasts have been thought to constitute the most abundant cell population in mammary tissues, difficulties in identifying mammary fibroblasts due to the lack of specific markers have been recognized [53]. More recently, carcinoma-associated fibroblasts (CAFs) have been delineated to four subgroups by large undertaking of mapping multiple cell markers using multicolor flow cytometry [38]. The major marker used to stratify CAF subgroups was α-SMA (alpha smooth muscle actin) as others have shown that fibroblast expression of α-SMA increases in wound healing, as well as stress conditions such as cancer [54]. Interestingly, our study results showed that α-SMA (encoded by *ACTA2*) was not expressed in fibroblasts (Cluster 3 and 6) but restricted to smooth muscle cells (Cluster 0) and basal epithelial cells (Cluster 7). These results suggest that normal fibroblasts in breast tissues may not express α-SMA at a readily detectable level, perhaps until they are activated by tumors. Costa et al. classified one of the CAF subgroups as α-SMA-negative and found that this subgroup designated CAF-S2 was detected only in luminal A breast cancer [38]. Concordantly, this CAF-S2 subgroup may correspond to Cluster 3 normal fibroblasts as both are α-SMA-negative and exclusively associated with luminal breast tumors. These indicate that CAFs found in non-luminal subtypes may undergo significant changes in gene expression depending on the tumor environment, specifically in hormone-negative tumors. Another fibroblast subset (Cluster 6) that we found in the normal breast did not match with any of TCGA breast tumors, suggesting that Cluster 6 fibroblasts may not directly contribute to breast carcinogenesis per se. However, it is formally possible that Cluster 6 fibroblasts could contribute to CAFs. Functional comparisons between Cluster 3 and Cluster 6 fibroblasts may reveal insight into the cell type origin of CAFs associated with non-luminal breast tumor subtypes.

### 3.5. Breast Cancer Subtypes Related to the Normal Cell Types in the Breast

The striking separation of the cell types matching to luminal (Cluster 2 and 3) vs. triple-negative breast cancer (Cluster 7 and 9) may indicates different cell origins. While it is consistent that luminal epithelial cells (Cluster 2) give rise to luminal tumors, the equal contribution of fibroblasts (Cluster 3) to luminal tumors warrants further investigation, especially related to therapy response. For example, “fibroblast” tumors may not have a robust response to hormone therapy but could respond to other growth factor receptor-targeted therapy.

Our data suggest that basal (Cluster 7) and other epithelial (Cluster 9) cell types mainly contribute to TNBC. TNBC, categorized because of the lack of the three receptors, ER, PR, and HER2, has further been subtyped according to gene expression signatures into either four intrinsic molecular subtypes [55] or six TNBC types [56]. The claudin-low subtype was added to the mix as a rare tumor type associated with stem-like properties [57]. Among these TNBC molecular subtypes, Cluster 9 tumors may be equivalent to the “basal-like” or BL1 subtype associated with poor prognosis and *BRCA* gene mutations. The basal-like subtype has been referred to almost synonymously with TNBC as the majority of TNBC tumors have been classified as basal-like [55]. However, our study showed that only 31% of TNBC tumors were Cluster 9 tumors, indicating likely differences between basal-like TNBC and Cluster 9 tumors. In addition, our data showed that more than 20% of TNBC tumors may arise from basal/myoepithelial Cluster 7 cells. It is unclear which previously defined TNBC subtype that Cluster 7 tumors may be equivalent to, since the current TNBC molecular subtyping system does not include a “basal/myoepithelial subtype”. These suggest that further investigation including functional assays is needed to clarify the cell type origin of TNBC.

HER2 tumors consisted of equal proportions of tumors matching to all four cell types (Clusters 2, 3, 7, and 9), attesting to the heterogeneous nature of the HER2 tumors, as demonstrated with the intrinsic molecular subtypes [39]. The immune cell (Cluster 8) signature was found in all subtypes of breast cancer, ranging from 21–23% in luminal subtypes to 43% in TNBC, highlighting the importance of immune components in cancer. The prevalence of Cluster 8 “immune” tumors in TNBC is consistent with the previous reports that TNBC tumors are found with high infiltrating immune cells and inflammatory gene expression [40,41]. It would be informative to correlate Cluster 8 tumors with response to therapy, especially to antibody agents that elicit a tumor immune response, such as trastuzumab and immune checkpoint inhibitors.

## 4. Methods and Materials

### 4.1. Breast Tissue Sample and Tissue Processing

The normal breast tissue was obtained from a patient who was diagnosed with ductal cell carcinoma in situ (DCIS) at the age of 59 years and opted for single mastectomy of the afflicted breast. The patient was postmenopausal and nulliparous at the time of surgery. The pathology examination post surgery confirmed a 2 cm single-focus DCIS. Normal tissue was taken from a quadrant uninvolved in the tumor from the mastectomy specimen. The sample was collected and provided by the Biobank Core Facility of St. Joseph’s Hospital and Medical Center at the Barrow Neurological Institute according to the approved Institutional Review Board protocol #PHXA-05TS038. The normal tissue of ~3 g in weight was dissected and divided into several portions, one of which was fixed in 4% paraformaldehyde (Ted Pella, Inc., Redding, CA) in phosphate-buffered saline (PBS) at 4 °C overnight followed by paraffin embedding and H&E staining using a BOND-MAX autostainer (Leica Microsystems, Wetzlar, Germany). The other portion of ~1.5 g of tissue was rinsed in cold PBS and mechanically minced to 0.1 mm^2^ in size using a scalpel. Tissue was frozen in 10% dimethyl sulfoxide (DMSO, Sigma, St. Louis, MO, USA) containing fetal bovine serum (FBS, Gibco, Thermo Fisher Scientific, Waltham, MA, USA) and stored at −80 °C.

### 4.2. Single-Cell Isolation and Sequencing

Single cells were isolated by first rapidly thawing the ~0.3 g of 0.1 mm^2^ size tissue bits in a 37 °C water bath. Tissue was washed with the Roswell Park Memorial Institute (RPMI) medium containing 2.5% fetal bovine serum (FBS, heat-inactivated, Gibco) and 10 units/mL penicillin–streptomycin antibiotics (Gibco) and subjected to sequential enzymatic digestion at 37 °C in a shaking platform in the same medium containing 2 mg/mL type IV collagenase (Sigma) for 40 min, 5 mg/mL dispase II (Roche, Basel, Switzerland) for 20 min, 0.25% trypsin (Gibco) for 5 min, and finally 1 mg/mL DNase I for 1 min. A minimum of two washes were carried out between the digests with the cold RPMI medium containing 2.5% FBS. Enzyme-dissociated tissue was run through a 70 μm cell strainer (Thermo Fisher Scientific). Single cells were washed and resuspended in cold PBS containing 1% bovine serum albumin (BSA, 10× Genomics). Approximately 15,000 single cells were transferred into the 10× reaction buffer (10× Genomics) and labeled using a GemCode Single-Cell platform to generate single-cell Gel Bead-in Emulsions (GEMs, 10× Genomics). Single-cell RNA-seq libraries were prepared using the Chromium™ Single-Cell 3’ Reagent Kit (10× Genomics) as per the manufacturer’s protocol, which involved barcoding (~750,000 barcodes) of reverse-transcribed complementary DNA (cDNA) and separately indexing each cell. Single-cell capture and library preparation were carried out using the GemCode Single-Cell 3′ Gel Bead-in Emulsions (GEMs, 10× Genomics) and Chromium™ Single-Cell 3′ v2 Reagent Kit following the manufacturer’s instruction for targeting 5000 single cells (10× Genomics). The resulting library constituted insert sizes with read lengths of >100 base pairs to a depth of 50,000 reads per cell using standard paired-end chemistry on the Illumina NextSeq 550 instrument (Illumina, San Diego, CA, USA).

### 4.3. Single-Cell RNA Sequencing Data Analysis

Raw sequencing data were preprocessed using Cell Ranger 2.1.0. In brief, the Cell Ranger mkfastq module was applied to generate fastq files by demultiplexing Chromium-prepared sequencing samples based on their barcodes. Thee fastq files were then input into the Cellranger count to generate unique molecular identifier (UMI) count data at a single-cell resolution. Further single-cell data analysis was conducted using R package Seurat (http://satijalab.org/seurat) [58,59]. In brief, we applied initial normalization where UMI counts for each cell were divided by the total number of counts and multiplied by 10,000. Cells expressing more than 6500 genes were filtered out for potential cell aggregates. Cells with a percentage of mitochondrial gene expression >0.1 were also filtered out for probable dead cells. These expression values were log-transformed before further downstream analyses. Principle component analysis, variable gene identification, shared nearest neighbor (SNN) clustering analysis, and t-distributed stochastic nearest neighbor embedding (t-SNE) visualization were then performed. In detail, the first 10 principle components were used for clustering analysis, and clusters were visualized with t-SNE mapping. Signature markers for each specific cluster were identified with function FindAllMarkers in Seurat against all remaining cells on the basis of the nonparametric Wilcoxon rank sum test. The top 100 significant markers with largest average log fold change were retained as the signature for each cluster. The differential gene expression heatmap was generated using the DoHeatmap function [59].

Pseudotemporal trajectory analysis was conducted using R package Monocle (version 2.8.0). Genes used for trajectory ordering were taken from the dispersion genes with normalization log_2_ mean expression >0.1. the DDRTree method was used for dimension reduction and cell ordering along the single-cell trajectories [60].

### 4.4. The Cancer Genome Atlas (TCGA) Data Analysis

TCGA Breast Invasive Carcinoma (TCGA-BRCA) RNAseq data (normalized log_2_ RSEM files) were downloaded from FireBrowser database (http://firebrowse.org), containing 1213 available patients. Corresponding patient clinical information including disease-free survival, overall survival, and breast cancer subtype designation were downloaded from cBioPortal (www.cbioportal.org) in TCGA Provisional Breast Invasive Carcinoma dataset, which were available for 1110 patients for analysis. Tumor subtypes were assigned according to the diagnostic data for ER and HER2 staining available in the dataset. Gene set variation analysis (GSVA) was performed to assign cell cluster gene signature scores per tumor ranging from −1 to +1 [34]. Each tumor was classified with the cell cluster number corresponding to the highest GSVA score.

### 4.5. Statistical Analysis

Kaplan–Meier survival plots were generated using GraphPad Prism (GraphPad Software, San Diego, CA, USA). The log-rank test was used to calculate the statistical significance between the survival curves. A *p*-value <0.05 was considered significant. All other statistical tests and figures were generated using various packages including stats, ggplot2, and ComplexHeatmap, in R 3.6.1. For gene expression levels, Student’s *t*-test was used to determine significance. A *p*-value <0.0001 was considered significant.

## 5. Conclusions

We presented an inclusive single-cell transcriptomic landscape of a postmenopausal normal breast tissue, showing three distinct epithelial and seven nonepithelial cell types. Among these, the epithelial, fibroblast, and immune cell signatures could collectively classify the majority of breast tumors, suggesting their contribution to breast carcinogenesis. Our findings may serve as a footprint as we move forward with comprehensive analyses of the cell types that constitute the normal breast. The knowledge gained by single-cell transcriptomic analyses here could provide insight into the healthy, pre-disease, and disease states, the therapy response, and the individual variations of the breast.

## Figures and Tables

**Figure 1 cancers-12-03639-f001:**
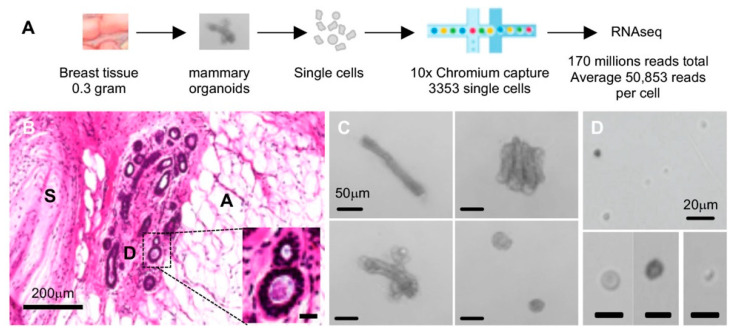
Normal breast tissue and single cell preparation. (**A**) Overview of the workflow; (**B**) formalin-fixed and paraffin embedded (FFPE) section stained with hematoxylin and eosin (H&E): D, mammary ducts; S, stroma; A, adipose tissue, scale bar represents 200 mm in length; inset contains a higher-magnification picture of the marked area representing mammary ducts, where the scale bar denotes 20 mm in length; (**C**) mammary organoids visualized after collagenase digest; scale bar denotes 50 mm in length; (**D**) single cells visualized following dispase II and trypsin sequential digests; scale bar represents 20 mm in length.

**Figure 2 cancers-12-03639-f002:**
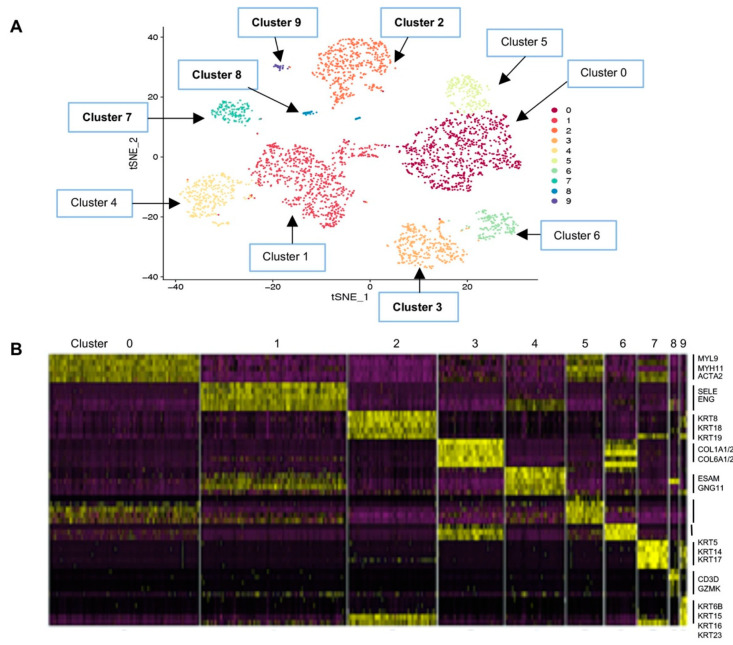
Human breast single-cell RNA sequencing (scRNAseq) clustering analysis. (**A**) t-Distributed stochastic neighbor embedding (t-SNE) map of scRNAseq data visualizing 10 cell clusters marked Clusters 0–10, in the order of the largest group of cells (Cluster 0) to the smallest (Cluster 9) listed in Table 1; (**B**) heat map of differentially expressed genes in each cluster. Some of the genes representing Clusters 1, 2, 3, 4, 5, 7, 8, and 9 are listed on the right side of the panel and in Table 1. Yellow to dark purple, high to low expression (the 100 genes differentially expressed in each cluster are listed in Appendix A).

**Figure 3 cancers-12-03639-f003:**
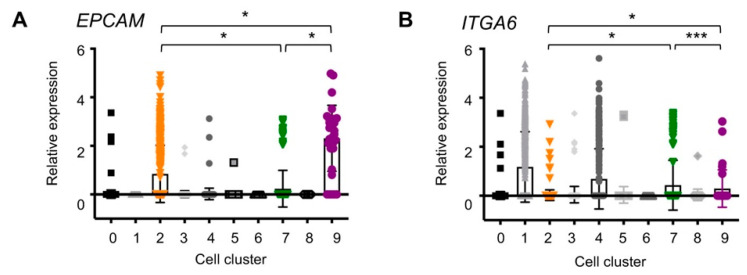
Epithelial cell marker expression in breast single-cell clusters (**A**) *EPCAM* (epithelial cell adhesion molecule, EpCAM) and (**B**) *ITGA6* (α6 integrin, Cd49f). Each dot represents a single cell; * *p* < 0.0001; *** *p* > 0.1. The median expression of the genes in each cluster is listed in Table 2.

**Figure 4 cancers-12-03639-f004:**
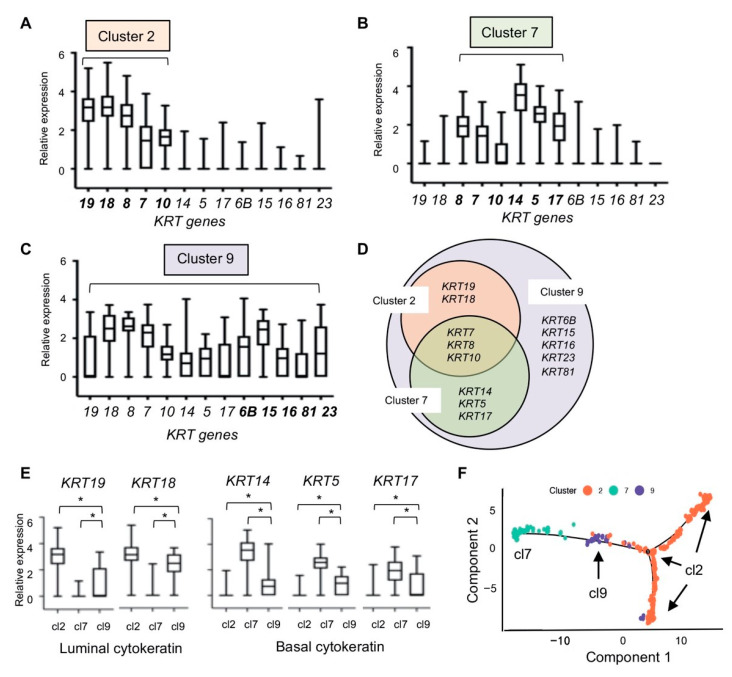
Cytokeratin gene expression in the three single cell clusters representing breast epithelial cells. (**A**) Cluster 2 (luminal); (**B**) Cluster 7 (basal); (**C**) Cluster 9; (**D**) Venn diagram of cytokeratin expression showing *KRT* genes expressed in Clusters 2, 7, and 9; (**E**) luminal and basal *KRT* expression in Clusters 2, 7, and 9. Box plots were generated using relative fold expression compared to the average expression of all cells; (**F**) pseudotemporal analysis. cl2, Cluster 2; cl7, Cluster 7; cl9, Cluster 9; * *p* < 0.0001 unpaired *t*-test.

**Figure 5 cancers-12-03639-f005:**
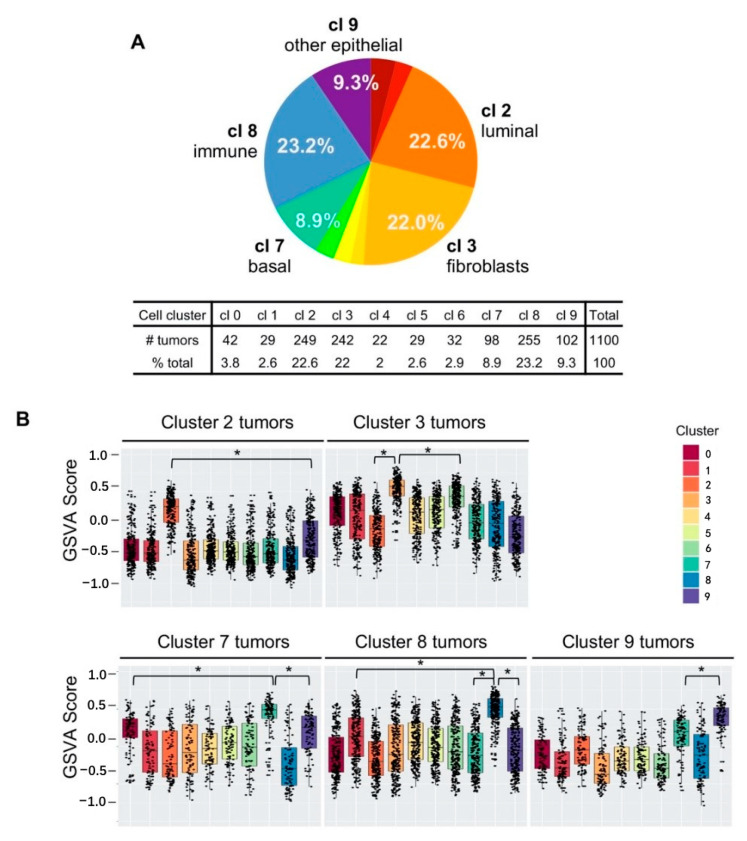
The gene signatures from five single cell clusters collectively classified 86% of breast tumors in The Cancer Genome Atlas (TCGA) dataset. (**A**) Percentages of breast tumors that matched to single cell clusters according to the gene set variation analysis (GSVA) scores. cl 2, Cluster 2; cl 3, Cluster 3; cl 7, Cluster 7; cl 8, Cluster 8; cl 9, Cluster 9; (**B**) cell cluster assignment showing the distribution of the assigned number. Cluster 2 tumors had the highest GSVA scores for the Cluster 2 gene signature compared to the other gene signatures (top left panel). Clusters 2, 3, 7, 8, and 9 are shown. Black dots represent a single tumor. All pairwise comparisons were significant; * *p* < 0.0001 was used to highlight the significance.

**Figure 6 cancers-12-03639-f006:**
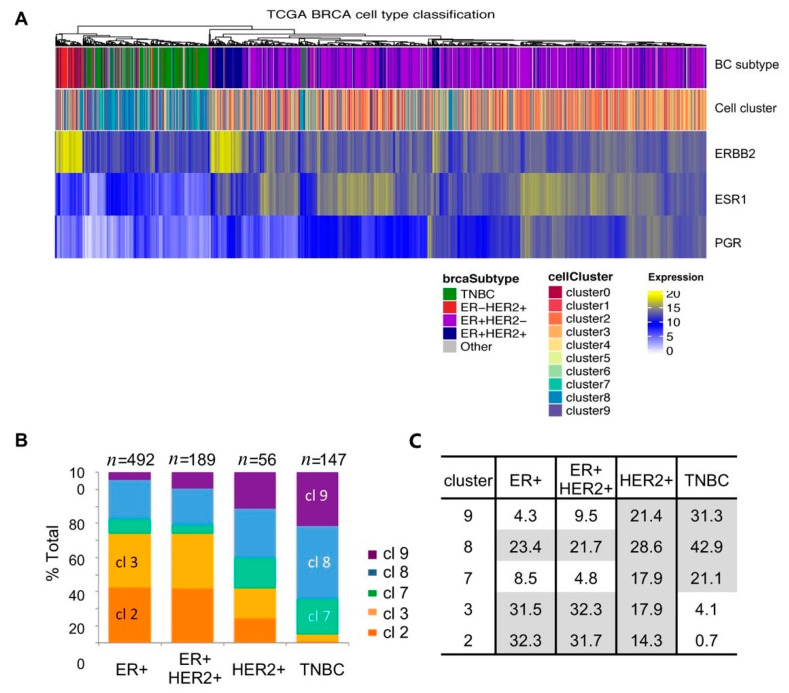
Single-cell clusters matching to the breast cancer subtypes (**A**) Heat map showing unsupervised clustering (dendrogram), matching to breast cancer subtype information in TCGA data (top panel), matching to the corresponding single-cell cluster number of the highest GSVA score (second panel), and expression levels of *ERBB2*, *ESR1*, and *PGR*. Color assignment is shown below the heat map; (**B**) Cluster 2 and 3 signatures were prevalent in the estrogen receptor-positive (ER^+^) tumors while Cluster 7 and 9 signatures were in the triple-negative breast cancer (TNBC) subtype. The percentage of each tumor type is listed in the table (**C**).

**Figure 7 cancers-12-03639-f007:**
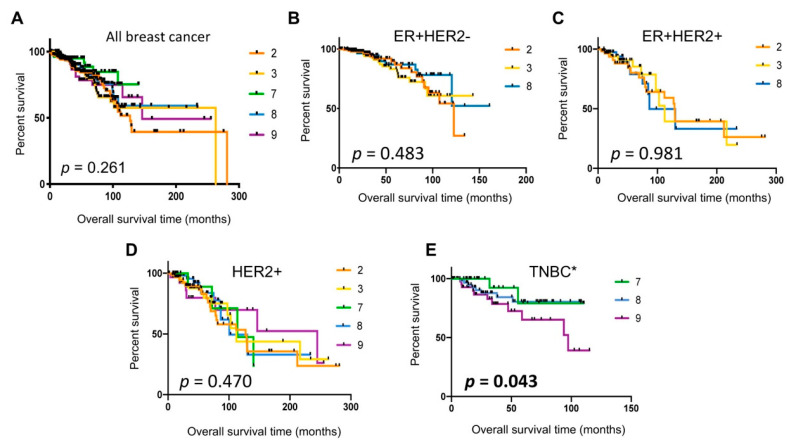
Kaplan–Meier survival analyses of TCGA breast cancer patient data using the single-cell cluster classification. (**A**) All breast cancer; (**B**) ER^+^HER2^−^; (**C**) ER^+^HER2^+^; (**D**) ER^−^HER2^+^; (**E**) TNBC. The *p*-values were determined by the log-rank test for trend. * Comparison between Cluster 7 and Cluster 9 tumors in TNBC, hazard ratio (HR) 3.493, 95% confidence interval (CI) 1.14–10.66.

**Table 1 cancers-12-03639-t001:** Cell cluster distribution and differentiating genes.

Cell Cluster	Number of Cells	% Total Cells	Genes	Proposed Cell Type
0	769	24.1	*ACTA2, MYH11, MYL9, MYLK, TAGLN*	Smooth muscle
1	749	23.5	*EMCN, ENG, PLVAP, SELE, SELP*	Endothelial
2	453	14.2	** KRT18, KRT19, KRT7, KRT8*	Luminal epithelial
3	336	10.5	*COL1A1/A2, COL3A1, COL6A1/A2/A3, ALDH1A1*	Stroma fibroblasts
4	305	9.6	*EDN1, EFNB2, ELTD1, ESAM*	Endothelial subset
5	188	5.9	Adipocyte markers: *APOE, CFD, STEAP4* Macrophage markers: *CCL2, CEBPD* Fibroblast markers: *COL4A1, COL6A3* EMT markers: *SNAI2, TGFB1*	Mesenchymal progenitor
6	161	5.0	*COL14A1, COL1A1/A2, COL6A2, ALDH1A1*	Stroma fibroblast subset
7	153	4.8	** KRT5, KRT14, KRT17, ACTA2, TAGLN*	Basal/myoepithelial
8	48	1.5	*C2, CD3D, CD7, CD8A, CD69, IL1B, TNF, GZMK*	Immune CTL
9	31	1.0	*EPCAM, * KRT6B, KRT15, KRT16, KRT81, KRT23*	Other epithelial
All	**3193**	**100.0%**		

* Cytokeratin gene expression in cell clusters 2, 7, and 9 (see Figure 3); EMT, epithelial–mesenchymal transition; CTL, cytotoxic T lymphocyte.

**Table 2 cancers-12-03639-t002:** *EPCAM* and *ITGA6* relative transcript amount in each cell cluster.

Cluster	0	1	2	3	4	5	6	7	8	9
*EPCAM*	0.01 ± 0.09	0	0.42 ± 0.59	0.01 ± 0.07	0.01 ± 0.12	0 ± 0.05	0	0.12 ± 0.38	0	1.15 ± 0.68
*ITGA6*	0.01 ± 0.08	0.59 ± 0.72	0.01 ± 0.11	0.02 ± 0.17	0.35 ± 0.62	0.02 ± 0.17	0	0.22 ± 0.51	0.02 ± 0.12	0.15 ± 0.38

The mean ± standard deviation of transcript amounts relative to the whole-tissue expression in each cell cluster; shaded boxes are the average expression exceeding 10% of normalized expression 1 (>0.1).

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
