# Peer review of "Single-Cell RNA Sequencing of a Postmenopausal Normal Breast Tissue Identifies Multiple Cell Types That Contribute to Breast Cancer"

_cancers, 2020, doi:10.3390/cancers12123639_

Round 1

Reviewer 1 Report

Peng and colleagues report results from a single cell analysis of normal postmenopausal breast tissue from a single patient. A major advantage of the study compared to prior studies is the use of total normal breast tissue for the single cell analyses versus pre-sorting for epithelial cells. This gives a valuable representation of diverse cell types within the tissue. The authors also argue that using tissue from a single sample rather than a pooled sample allows for less variability. However, it also limits the interpretations possible as the analysis of only one sample could skew results and interpretations. The authors identify three different epithelial populations and seven different non-epithelial populations. They then compare the gene signatures of each population to gene expression profiles of breast tumors in the TCGA dataset. Overall this is a very interesting and valuable study. Two major limitations are in drawing conclusions from the single sample as mentioned above as well as the overinterpretation of the normal cell data alignment with human breast cancers. The conclusion that “five cell types significantly contributed to breast cancer” overinterprets the findings.

Specific Comments:

  • Are the novel endothelial (cluster 4) and fibroblast (cluster 6) populations identified thought to be different cell states? It might be interesting to perform pseudotime analyses.

  • A pseudotime analysis would also be interesting if the origin is set at cluster 5 to test the hypothesis that these cells are mesenchymal progenitors. The lineage relationship of the epithelial clusters might also be of interest, particularly for understanding cluster 9 identity.

  • Figure 5 data are interesting but somewhat difficult to interpret given that the comparison is being made between gene expression of normal breast cell populations with that of breast tumor cells. For example, would the correlation with fibroblast clusters (referred to as “stroma” by the authors) represent mesenchymal type tumors?

  • Throughout the manuscript, the authors interpret their alignment with normal breast cell clusters to breast cancer subtypes as indicative that these normal cells give rise or are the origin of specific cancer subtypes. This represents an overinterpretation of their data and needs to be tempered.

  • Similarly, the authors need to acknowledge the limitation of the comparisons using only one sample along with what they note as a strength of the data from not pooling multiple samples.

  • There is little mention of the data in the literature indicating that luminal progenitor cells are the cell of origin for many TNBCs. If cluster 9 is a distinct population from the luminal progenitor population as the authors argue, why is it they have not identified a luminal progenitor population? They also argue the luminal progenitor population would be expected to be a larger component but would that be true in postmenopausal breast tissue? This needs to be better acknowledged and analyzed.

  • Minor Comment: Figure 2B – the cluster numbers appear shifted to the left on the heat map such that they do not align with the actual cluster gene expression particularly on the far right.

Author Response

Response to Reviewer 1 Comments

We thank the reviewer for the insightful and instructive comments on our manuscript. We revised manuscript in response to the reviewers’ comments and recommendations. The revised portions are highlighted using track changes in the manuscript.

We believe that we have addressed the reviewers’ concerns and, in doing so, have improved the manuscript. Please let me know if you have any questions or additional concerns.

The revision addresses the reviewers’ comments (in black) and our responses (in red) as follows:

Peng and colleagues report results from a single cell analysis of normal postmenopausal breast tissue from a single patient. A major advantage of the study compared to prior studies is the use of total normal breast tissue for the single cell analyses versus pre-sorting for epithelial cells. This gives a valuable representation of diverse cell types within the tissue. The authors also argue that using tissue from a single sample rather than a pooled sample allows for less variability. However, it also limits the interpretations possible as the analysis of only one sample could skew results and interpretations. The authors identify three different epithelial populations and seven different non-epithelial populations. They then compare the gene signatures of each population to gene expression profiles of breast tumors in the TCGA dataset. Overall this is a very interesting and valuable study. Two major limitations are in drawing conclusions from the single sample as mentioned above as well as the overinterpretation of the normal cell data alignment with human breast cancers. The conclusion that “five cell types significantly contributed to breast cancer” overinterprets the findings.

Specific Comments:

  1. Are the novel endothelial (cluster 4) and fibroblast (cluster 6) populations identified thought to be different cell states? It might be interesting to perform pseudotime analyses.

Thank you for the suggestion. We performed a pseudotime analysis and found that Cluster 1 appears to precede Cluster 4 while Cluster 6 preceded Cluster 3 on the contiguous trajectory timeline, suggesting potential differentiation lineage relationships between the respective cell subtypes. We included these findings in the text (lines 210-213) and presented the pseudotime analyses as Supplementary Figure S1.

  1. A pseudotime analysis would also be interesting if the origin is set at cluster 5 to test the hypothesis that these cells are mesenchymal progenitors. The lineage relationship of the epithelial clusters might also be of interest, particularly for understanding cluster 9 identity.

The pseudotime analysis presented in Supplementary figure S1 was done with Cluster 5 at the “root,” which showed that Cluster 5 may indeed represent mesenchymal progenitors potentially differentiating to smooth muscle (Cluster 1), fibroblasts (Cluster 3 and 6), and endothelial cells (Cluster 1 and 4). We added this results in the text (lines 219-225) and Supplementary figure S1.

Thank you for your perceptive suggestion of performing a pseudotime analysis of the epithelial cell types (Cluster 2, 7, and 9), which revealed that Cluster 9 in fact mapped to the origin on the differentiation timeline, supportive of our proposal that Cluster 9 may represent bipotent progenitors. We included this pseudotime analysis as Figure 4F and described the results in the text (lines 301-306).

  1. Figure 5 data are interesting but somewhat difficult to interpret given that the comparison is being made between gene expression of normal breast cell populations with that of breast tumor cells. For example, would the correlation with fibroblast clusters (referred to as “stroma” by the authors) represent mesenchymal type tumors?

We had also found it perplexing that the fibroblast (Cluster 3) signature matched exclusively to luminal subtype tumors instead of TNBC including mesenchymal type tumors that has been shown with aggressive CAF markers. However, we found that our results were consistent with Costa et al (Cancer Cell, 2018), of which extensive characterization of CAFs showed that one of the CAF subsets associated specifically with luminal tumors. We believe this CAF subset (CAF-S2) matches to Cluster 3, normal fibroblasts, in our study. This leaves the question of other CAF subsets’ origins. This point was discussed in Discussion section 3.4, which included a plausible explanation that “normal fibroblasts” may undergo extensive gene expression changes in TNBC and HER2 tumors.

Likewise, previous studies have classified mesenchymal type tumors (also referred as claudin-low tumors elsewhere) as a TNBC subset. Cluster 5 (mesenchymal progenitor) gene signature did not match to any subtype of breast tumors including mesenchymal tumors. However, it is formally possible that Cluster 5 cells undergo extensive gene expression changes in tumors. Experimental investigation using Cluster 5 cells may address this interesting point, but is beyond the scope of this manuscript.

  1. Throughout the manuscript, the authors interpret their alignment with normal breast cell clusters to breast cancer subtypes as indicative that these normal cells give rise or are the origin of specific cancer subtypes. This represents an overinterpretation of their data and needs to be tempered.

We agree with the reviewer’s comment. Although we are enthusiastic about the cell origin idea, we agree it is overreaching with our data at hand. We replaced “give rise” to “have similar gene expression profiles” or “may contribute to” throughout the text.

  1. Similarly, the authors need to acknowledge the limitation of the comparisons using only one sample along with what they note as a strength of the data from not pooling multiple samples.

We agree with the reviewer’s comment and have explicitly acknowledged the limit of using one sample in the text (lines 502-503).

  1. There is little mention of the data in the literature indicating that luminal progenitor cells are the cell of origin for many TNBCs. If cluster 9 is a distinct population from the luminal progenitor population as the authors argue, why is it they have not identified a luminal progenitor population? They also argue the luminal progenitor population would be expected to be a larger component but would that be true in postmenopausal breast tissue? This needs to be better acknowledged and analyzed.

We have referenced Lim et al (Nat Med 2009) and Prat and Perou (Nat Med 2009) as the source literature for luminal progenitors as the cell origin of a TNBC subset.

The reviewer’s question of why we have not identified luminal progenitor population in our study is a perceptive one. The psudotime analysis of epithelial cell clusters that the reviewer suggested in #2 comments address this point – Cluster 2 (luminal) branched out to two lineages in the differentiation trajectory, aligning well with the Nguyen et al (Nature Comm 2018) trajectory with luminal progenitors. This indicated that our Cluster 2 appeared to contain luminal progenitors as well as the other luminal cell states. Cluster 9 mapping to the root in the trajectory is supportive of our proposal that Cluster 9 represents rare bi-potent progenitor cells. By incorporation these points, we improved Discussion section 3.2 accordingly.

  1. Minor Comment: Figure 2B – the cluster numbers appear shifted to the left on the heat map such that they do not align with the actual cluster gene expression particularly on the far right.

This has been corrected. Thank you!

Reviewer 2 Report

The authors present a manuscript looking at the single-cell RNA sequencing of a postmenopausal normal breast tissue isolated from the mastectomy of a patient with ductal cell carcinoma in situ. The authors reported multiple clusters of cell types that contribute to breast cancer. Despite the potential for this being a relevant study, it suffers from minor issues that need to be addressed to improve its overall quality. Please find my minor comments below;

  1. As the cells were isolated from postmenopausal normal breast tissue, the TCGA-BRCA RNAseq data used for comparison were from postmenopausal or pre?
  2. There is some minor discussion about stem cell features in cluster 5, it would be interesting to see the status of stemness or pluripotency markers in breast cancer clusters.
  3. The author should also include a graphical overview of the workflow.

Author Response

Response to Reviewer 2 Comments

We thank the reviewer for the insightful and instructive comments on our manuscript. We revised manuscript in response to the reviewers’ comments and recommendations. The revised portions are highlighted using track changes in the manuscript.

We believe that we have addressed the reviewers’ concerns and, in doing so, have improved the manuscript. Please let me know if you have any questions or additional concerns.

The revision addresses the reviewers’ comments (in black) and our responses (in red) as follows:

The authors present a manuscript looking at the single-cell RNA sequencing of a postmenopausal normal breast tissue isolated from the mastectomy of a patient with ductal cell carcinoma in situ. The authors reported multiple clusters of cell types that contribute to breast cancer. Despite the potential for this being a relevant study, it suffers from minor issues that need to be addressed to improve its overall quality. Please find my minor comments below;

  1. As the cells were isolated from postmenopausal normal breast tissue, the TCGA-BRCA RNAseq data used for comparison were from postmenopausal or pre?

This is a good point. We conducted the same analysis using postmenopausal tumors only from the TCGA dataset. The results showed the same distribution of the Cluster signatures, indicating that postmenopausal normal cell types can represent the majority of breast tumors regardless of the patient menopausal status. We added the data as Supplementary figure S2 and stated the results in the text (lines 461-466).

  1. There is some minor discussion about stem cell features in cluster 5, it would be interesting to see the status of stemness or pluripotency markers in breast cancer clusters.

Cluster 5 signature did not match to breast tumors in the TCGA data set. We added an additional pseudotime analysis of Cluster 5 in Supplementary Figure S1, which showed differentiation trajectories of Cluster 5 to various non-epithelial cell types. While this is supportive of our proposal that Cluster 5 may represent mesenchymal progenitors, the signature did not match to “mesenchymal tumors,” suggesting that either Cluster 5 cells do not contribute to breast tumors or undergo extensive gene expression changes that our data do not recognize as the same cell type.

The pluripotency markers (e.g. SOX2, KLF4, MYC, NANOG, OCT4) did not yield any cohesive cluster/tumor distribution. Stemness signature analysis would be interesting but we believe is beyond the scope of this manuscript.

  1. The author should also include a graphical overview of the workflow.

 It is added as Figure 1A. Thank you for the suggestion.

Round 2

Reviewer 1 Report

The authors have done a thoughtful and careful job of addressing my prior concerns and have provided a nicely revised manuscript with new analyses and revised conclusions.

Reviewer 2 Report

The authors satisfactorily responded to my comments.